# Antimicrobial Stewardship Programs: A Review of Strategies to Avoid Polymyxins and Carbapenems Misuse in Low Middle-Income Countries

**DOI:** 10.3390/antibiotics11030378

**Published:** 2022-03-12

**Authors:** Fabrício Rodrigues Torres de Carvalho, João Paulo Telles, Felipe Francisco Bodan Tuon, Roberto Rabello Filho, Pedro Caruso, Thiago Domingos Correa

**Affiliations:** 1Intensive Care Unit, Hospital Israelita Albert Einstein, São Paulo 05652-900, SP, Brazil; roberto.rabello@einstein.br (R.R.F.); thiago.correa@einstein.br (T.D.C.); 2Intensive Care Unit, AC Camargo Cancer Center, São Paulo 01525-001, SP, Brazil; pedcaruso@gmail.com; 3Department of Infectious Diseases, AC Camargo Cancer Center, São Paulo 01525-001, SP, Brazil; 4School of Medicine, Pontifical Catholic University, Curitiba 80215-901, PR, Brazil; felipe.tuon@pucpr.br; 5Department of Infectious Diseases, Hospital Universitario Evangelico Mackenzie, Curitiba 80730-420, PR, Brazil

**Keywords:** antimicrobial stewardship, antimicrobial resistance, carbapenems, polymyxins, procalcitonin

## Abstract

Antibiotics misuse and overuse are concerning issues worldwide, especially in low middle-income countries. These practices contribute to the increasing rates of antimicrobial resistance. One efficient strategy to avoid them is antimicrobial stewardship programs. In this review, we focus on the possible approaches to spare the prescription of polymyxins and carbapenems for the treatment of *Acinetobacter baumannii*, carbapenem-resistant *Enterobacterales*, and *Pseudomonas aeruginosas* infections. Additionally, we highlight how to implement cumulative antibiograms and biomarkers to a sooner de-escalation of antibiotics.

## 1. Introduction

Antimicrobial resistance (AMR) has been discussed by the World Health Organization (WHO) since 2015 through the WHO Global Antimicrobial Resistance and Use Surveillance System (GLASS). In June 2021, WHO reported the fifth data from GLASS, accounting for more than a hundred countries [1]. The rates of bloodstream infections due to *Escherichia coli* resistant to third-generation cephalosporin and *Staphylococcus aureus* resistant to methicillin were higher in low middle-income countries compared with high-income countries (58% versus 17% and 33% versus 15%, respectively) [1]. Moreover, around 65% of all *Acinetobacter* spp. classified as a bloodstream infection were carbapenem-resistant [1]. In Europe, 30% of the *Acinetobacter* spp. and 12% of the *Pseudomonas aeruginosa* are resistant to multiple antimicrobial options (i.e., carbapenems, fluorquinolones, and aminoglycosides) (https://www.ecdc.europa.eu/sites/default/files/documents/surveillance-antimicrobial-resistance-Europe-2019.pdf; accessed on 1 May 2021). Additionally, it is expected that 10 million deaths each year will be due to drug-resistant bacteria in 2050 [2].

Antimicrobial stewardship programs (ASP) play an important role in preventing the increase of AMR [3]. However, a meta-analysis demonstrated that there is heterogeinity in the included studies, and preventing the increase of AMR is better achieved when the infection prevention and control department develops activities along with the ASP team [4]. Additionally, there are different strategies of ASPs, such as prospective audit and feedback, preauthorization, or education programs. [5]. Nevertheless, it is recommended that all core elements such as education programs, along with antimicrobial restriction by prospective audit and feedback, are used together to improve outcomes (e.g., decreasing AMR and *Clostridium difficille* infection). It is important to highlight that, without hospital leadership commitment, ASP efforts may not be transformed into results. In different regions around the world, ASP has been demonstrated to be effective in decreasing AMR, *Clostridioides difficille* infection, and hospital costs [6,7,8,9].

There is evidence that “hit hard and hit fast” would be the best approach favoring the host instead of pathogen during the dynamics of an infection process, as previously thought by Paul Ehrlich in the beginning of the 20th century [10]. Nevertheless, it was never stated that the strength of the strategy would be related to its amplitude. Unfortunately, medical clinical practice commonly assumes that the harder you hit, the broader antimicrobial therapy you must prescribe. Antimicrobial de-escalation aims to attenuate the consequences of broad antimicrobial usage. However, as previously stated, this strategy should not be synonymous with safeguarding against inappropriate antibiotic prescription [11]. As diagnostic stewardship (e.g., a cumulative antibiogram analysis) favors a better understanding of the microbiological results, it allows a sooner de-escalation of the antibiotics, thus decreasing the antimicrobial selective pressure [12].

The aim of this review is to highlight the alternatives to spare polymyxin and carbapenem usage to treat *Acinetobacter baumannii*, carbapenem-resistant *Enterobacterales*, and *Pseudomonas aeruginosas* infections. Moreover, we also describe how to implement cumulative antibiograms and biomarkers for a sooner de-escalation.

## 2. Alternative Polymyxin-Sparing Regimens

Polymyxin-based therapy remains as the gold standard for carbapenem-resistant *Acinetobacter baumannii* (CRAB) and extensively drug-resistant (XDR) *Pseudomonas aeruginosa*. However, these drugs have important disadvantages. Pharmacokinetics and pharmacodynamics data are scarce, susceptibility tests are based on broth microdilution, and side effects are common [13,14]. Additionally, some regions of the globe have suffered polymyxin shortage during the COVID-19 pandemic, with a 400% increase in price (e.g., Brazil) [15] (https://oglobo.globo.com/saude/apos-vencerem-covid-19-pacientes-morrem-de-infeccao-hospitalar-por-falta-de-antibiotico-25123631; accessed on 1 November 2021) [16]. Therefore, alternative therapies are needed to avoid side effects and improve cost-effectiveness.

### 2.1. Acinetobacter Baumannii

#### 2.1.1. Major Surveillance Data from Acinetobacter Baumannii Resistance Rates

Considering the last data from *Acinetobacter baumannii* susceptibility, the results may differ between regions. For instance, data on multidrug-resistant (MDR) *Acinetobacter baumannii* from the United States of America (2014–2018) found that the MIC (minimum inhibitory concentration)_50__–90_ of levofloxacin, amikacin, cefepime, and meropenem were 4 mg/L, 8–32 mg/L, 16 mg/L, and 32 mg/L, respectively [17]. However, minocycline MIC_50__–90_ was 2–8 mg/L, while more than 70% of the isolates were minocycline-susceptible, according to the Clinical and Laboratorial Standards Institute [17]. Even in extensively drug-resistant *Acinetobacter baumannii* strains (XDR), minocycline was the second-most active drug, with a 67% susceptibility, and polymyxin was the first, with 86% [17]. Between 2013 and 2016, the XDR *Acinetobacter calcoacetius–Acinetobacter baumannii* complex accounted for 86% of the isolates from Latin America, whereas the percentange was 40% in North America [18]. For these reasons, it is important to know the local susceptibility patterns for a better therapy choice.

#### 2.1.2. Data from Alternative Therapies for Polymyxin-Sparing Regimens to CRAB

Previous studies did not evaluate the outcomes of patients treated with polymyxins compared with alternative antibiotics. Nevertheless, cohort studies have demonstrated encouraging results of minocycline/doxycycline, sulfamethoxazole–trimethoprim (S–T), and sulbactam-based therapy [19,20,21,22]. The tetracyclines (monotherapy or combined-based regimens) used depicted > 70% clinical success, including patients with ventilator-associated pneumonia (VAP) [19,20,21].

Despite these results, caution should be taken, mainly to avoid tetracyclines monotherapy in patients with bloodstream infections, as these drugs present a large volume of distribution and low serum concentrations [23]. Sulfamethoxazole–trimethoprim is also an option when the susceptibility is known. A retrospective cohort found no differences in 30-day mortality between colistin versus S–T in monotherapy (38% versus 25%, respectively, *p* > 0.05) [23]. The major concerns of S–T use in critically ill patients are (i) the shortage of intravenous formulation on the market and (ii) the large volume of distribution, with a higher concentration in tissues [24]. Data on the total S–T serum concentrations showed that high-dose regimens (i.e., 2400 mg of sulfamethoxazole per day) can achieve serum concentrations higher than the MIC values [25]. However, free concentrations should be further investigated.

Sulbactam-based therapy may also be an option to spare polymyxins. The dosage regimen of sulbactam in critically ill patients should be at least 3 g q8h (1-h infusion) to achieve MICs as high as 8–16 mg/L [26]. A prospective cohort study compared sulbactam versus polymyxin in patients with VAP and found similar mortality rates between the regimens [27]. Unfortunately, MIC_50__–90_ of 8–16 mg/L for sulbactam in *Acinetobacter calcoacetius–A. baumannii* isolates were only found in North America, whereas MIC_50_ > 16 mg/L were usually found worldwide [18,28].

### 2.2. Carbapenem-Resistant Enterobacterales (CRE)

#### 2.2.1. Major Surveillance Data from Carbapenem-Resistant Enterobacterales

There has been an increased incidence of carbapenem-resistant *Enterobacterales* worldwide, mainly due to *Klebsiella pneumoniae* [29]. More than 50% of CRE isolates carry the *bla*_KPC_ gene, and 13% of carbapenem resistance is due to metallo-beta-lactamase [29]. Among CRE isolates, the MIC_50__–90_ of tigecycline, polymyxin, and amikacin were 0.5–2 mg/L, 0.25–8 mg/L, and 16–32 mg/L, respectively [30]. Thus, the major therapeutic option according to these results should be tigecycline. Almost 90% of these CRE isolates were from North America and Europe [30]. In Latin American, studies have demonstrated similar tigecycline MIC distributions in carbapenems producing *Enterobacterales* but a lower MIC_50_ of amikacin, 4 mg/L [31]. Unfortunately, the major increase of CRE in the past years occurred in Latin America. This region has poor access to the new therapeutic options such as ceftazidime–avibactam and meropenem–vaborbactam. Therefore, a co-resistance to carbapenem and polymyxin is usual there [29,32,33].

#### 2.2.2. Data from Alternative Therapies for Polymyxin-Sparing Regimens to CRE

Although tigecycline MIC distribution is relatively low, caution is needed regarding the choice of therapy. Tigecycline 50 mg q12h commonly does not achieve enough serum concentration to treat patients with severe infections, and a higher dosage regimen such as 100 mg q12h is needed [34]. In addition, a combination therapy with CRE is commonly used, and there is evidence of antagonism (in vivo and in vitro) between tigecycline and meropenem [35].

Aminoglycosides remain as an important option to treat CRE. Nevertheless, monotherapy is still a concern, depending upon the site of infection. Considering its pharmacokinetics, the use of high doses is needed in critically ill patients to achieve suitable serum concentrations to treat CRE infections (e.g., amikacin MIC of 8 mg/L and gentamicin MIC of 2 mg/L) [36,37], while epithelial lining fluid may still present low concentrations [38]. However, the methods to evaluate pulmonary concentrations are not well-established [39]. Additionally, polymyxin pulmonary penetration is poor, and a combined therapy with meropenem did not demonstrate better outcomes [39,40]. Therefore, aminoglycosides monotherapy to treat lung infections needs further investigation.

New therapeutic options for CRE are based on new beta-lactamase inhibitors. Therapies based on avibactam, relebactam, and vaborbactam are the first treatment options if these MDR pathogens are found [41]. Unfortunately, these options are usually not available in low middle-income countries.

## 3. Alternative Carbapenem-Sparing Regimens

### 3.1. Major Surveillence Data from Cephalosporin-Resistant Enterobacterales

Third-generation cephalosporin-resistant bacteria have increased continually since the 1990 decade. During this period, *Enterobacterales* with third-generation cephalosporin resistance accounted for 10–15%, while, in 2013–2016, it increased up to 20–25% [29]. Some countries such as Belgium, France, Germany, and Ireland more than tripled their third-generation cephalosporin resistance rates [29]. Additionally, in Latin America, countries such as Brazil, Argentina, Mexico, Chile, and Honduras also convereged on an important increase of ESBL among Enterobacteriales [42].

These resistant patterns are caused by an increase in Extended-Spectrum Beta-Lactamases (ESBLs) such as CTX-Ms [43,44]. Consequentially, carbapenem prescription has increased and, as a result, the bacteria’s resistance to it. Therefore, a sparing carbapenem regimen therapy plays an important role in breaking the Gram-negative bacilli resistance pathway. However, alternative therapies such as quinolones, aminoglycosides, and group 1 carbapenem (i.e., ertapenem) have only been studied in retrospective cohorts.

### 3.2. Data from Alternative Therapies for Carbapenem-Sparing Regimens to Cephalosporin-Resistant Enterobacterales

For a long time, quinolones were used as a carbapenem-sparing option for infections caused by ESBL or AmpC isolates. However, the increase of quinolones resistance in ESBL-positive bacteria has put them aside, at least as an empirical option. In North America, quinolone susceptibility patterns from urinary isolates of ESBL-positive *E. coli* vary only between 7 and 26% (i.e., Ciprofloxacin and Levofloxacin) [45]. Data from Asia and Pacific countries have demonstrated a 20–30% quinolone susceptibility in ESBL-positive *E. coli* also from urinary tract infections [46]. Intra-abdominal infections may present higher quinolone patterns of susceptibility. As an example, in Asia and Pacific countries, ESBL-positive *K. pneumoniae* presented 57% susceptibility [47]. Therefore, despite the possible co-resistance of third-generation cephalosporins and quinolones, ASP should be aware of the quinolone local resistance rate and evaluate its de-escalation as soon as possible.

Beta-lactam/beta-lactamases inhibitors (BL/BLI) such as piperacillin–tazobactam were commonly used in infections caused by ESBL and Amp-C isolates. The MERINO trial, which included only bloodstream infections, compared meropenem with piperacillin–tazobactam and found a higher 30-day mortality in the piperacillin–tazobactam patient subgroup [48]. The results of the study have been questioned, as methodological problems have been noted, such as the drug infusion time, false–positive sensitivity to piperacillin–tazobactam, and mortality more frequent in patients with cancer [49]. Thus, BL/BLI is still open to discussion, and new trials are ongoing [44]. Despite that, BL/BLI, including amoxicillin–clavulanate, may be interesting options for urinary tract infections [50]. Caution is needed to treat severe infections due to the lack of amoxicillin–clavulanate pharmacokinetics and pharmacodynamics data [51].

Besides the use of aminoglycosides for CRE infection, amikacin and gentamicin also have a central role in the treatment of ESBL- and AmpC-positive isolates. Studies have demonstrated a high susceptibility pattern around 90–95% of ESBL-positive bacteria [45,47]. However, as stated previously, it is necessary to be careful to interpret the clinically treatable MICs of amikacin and gentamicin and to consider the severity of infection (e.g., higher dosage regimen in critically ill patients). Renal toxicity has been evaluated, and in studies using the propensity score matching in different populations, the results demonstrated that septic patients treated with meropenem or aminoglycosides presented the same proportition of acute kidney injury [52].

Ertapenem may be an option to spare meropenem and imipenem. A recent meta-analysis concluded that ertapenem prescription does not increase carbapenem resistance either in *Pseudomonas aeruginosas* or in Enterobacterales [53]. The dosage of ertapenem of 1 g q24h is an option for earlier hospital discharge, presenting good cost-effectiveness [54]. Thus, despite being a carbapenem, the antimicrobial stewardship program can use ertapenem to spare carbapenem group 2.

## 4. Cumulative Antibiogram

In recent years, the concept of cumulative antibiogram has been applied to the decision-making process for antibiotic selection before the final antimicrobial susceptibility testing results. To define the best choice of antibiotics, the historical susceptibility profile of the hospital or the community must be acknowledged. In case a pathogen is identified, an alternative that best fits the profile of the patient should be sought. For the construction of the cumulative antibiogram, some premises defined by the Clinical Laboratory Standards Institution (CLSI) have been suggested and are contained in the document CLSI M39-A2 [55]. An example of a cumulative antibiogram is detailed in Figure 1.

The microbiological tests will support medical judgments and the decision-making process. This allows for the most appropriate antibiotic use, especially in the current era of multidrug resistance.

In setting up the cumulative antibiogram of Gram-negative bacilli, it is necessary to find the best scheme for the main infection site (e.g., bacteremia, respiratory tract, and urinary tract). It is also important to remember that the combined therapy is guided to extend the spectrum, not for synergism. Ideally, during the cumulative antibiogram, quinolones and carbapenem should be avoided to decrease the selective pressure for multidrug-resistant bacteria. In this case, combinations with aminoglycosides can be an option [6]. A previous study that evaluated the use of quinolones and aminoglycosides by a cumulative antibiogram as a carbapenem-sparing strategy showed a reduction of carbapenem resistance after one year [6]. Different from expected, after one year of increased use of aminoglycosides, the susceptibility of this antimicrobial class increased, which is justified by a reduction in the carbapenem efflux system in some MDR microrganisms.

In spite of a useful, cumulative antibiogram, the laboratorial and ASP team needs constant communication and larger files (e.g., Excel) with all the culture and susceptibility tests results. Lastly, a cumulative antibiogram analysis must be done carefully in order to avoid misinterpretation between the different antimicrobial options tested in one bacteria species and not in another.

## 5. Stewardship Based on Biomarkers

A distinction between viral and bacterial infections is challenging, particularly during the initial clinical presentation. As an example, in COVID-19, the hyperinflammatory phase of the late phase is difficult to differentiate from a secondary bacterial infection [56]. Therefore, many patients tend to mistakenly receive antibiotic therapy.

From a clinical perspective, biomarkers may help in the diagnosis and prognosis. The ideal biomarker should present fast kinetics, high sensitivity, and specificity. Regularly, the most used markers for the follow-up of patients with bacterial infections are the white blood cell count and the C-reactive protein (CPR). Additionally, procalcitonin (PCT) has been widely studied for this purpose [57,58].

CRP is one of many nonspecific acute-phase reactants that are elevated during an inflammatory process. Since the CRP response to an inflammatory process is nonspecific, many clinicians have not adopted its use as a predictive and prognostic test in intensive care medicine [59]. Furthermore, the role of CRP as a predictor of infection, instead of inflammation, has become even more controversial since the introduction of PCT as a test in this regard. Comparing CRP with other inflammatory markers (specially PCT) can be difficult because of their different kinetics. The overall evidence suggests that PCT has much faster kinetics, both in its onset and offset, and may also be more specific than the CRP in diagnosing some infections [58,59,60]. Since the CRP test is widely available and relatively cheaper than PCT, it is likely to be widely used in many institutions, including in low-income countries. Although CRP is more widely used, evidence suggests that PCT could be more useful regarding decisions of starting and discontinuing antibiotics and could help decrease antibiotic consumption and mortality in the ICU [58,60,61].

In some previous studies, PCT has proven useful in the early detection of bacterial infections, lower respiratory tract infections being the most widely studied [58,62]. It has been shown that PCT levels have greater sensitivity and specificity than CPR levels for detecting and following up patients with bacterial infections [58]. A possible explanation is that viral infections elevate the interferon-gamma (IFN-γ) serum levels, reducing the upregulation of PCT, which justifies the high specificity of PCT for distinguishing viral from bacterial infections [63,64].

A meta-analysis, including a total of 3244 patients from 30 high-quality studies, found that PCT presented a ROC curve of 0.85 (95% CI: 0.81–0.88) to distinguish sepsis from a systemic inflammatory response syndrome (SIRS) of noninfectious origins, an excellent result [65]. As in one of the previously cited studies [58], in this meta-analysis, the majority of the included studies were done in the ICU, while 60% of patients were diagnosed with sepsis [65]. Nevertheless, in spite of these promising results, another meta-analysis in which the majority of patients were not diagnosed with sepsis nor needed ICU support, the ROC curve was lower, 0.73 (95% CI: 0.69–0.76) [63]. Therefore, there is a possible difference of the bacterial sepsis ROC curve between critically ill and noncritically ill patients, demonstrating that the clinical decision should not be based only on PCT levels.

Recently, an international guideline was published recommending the use of PCT levels in intensive care patients to estimate the bacterial infection probability [66]. The cut-off value that contemplates a lower probability of bacterial infection was set at <0.5, but serial measurements of PCT in 24–48 h can be necessary [66]. Furthermore, PCT-based antibiotic stewardship protocols should not be applied to patients with chronic infections, such as osteomyelitis, abscess, and endocarditis [66]. In addition, a more recent meta-analysis focusing on a PCT-guided antibiotic in septic patients with different sites of infection included 11 randomized trials with 4482 patients and demonstrated lower mortality with PCT-guided therapy [61]. Moreover, the PCT-guided patient group was associated with a shorter treatment duration of antibiotics [61]. These results reaffirm the safety and effectiveness of this biomarker use, leading to adequate diagnosis and treatment, the cornerstones of antibiotic stewardship.

There is still a lack of data regarding using biomarkers in ASP with the objective of decreasing the consumption of carbapenems and polymyxins. Nonetheless, the current evidence for starting and stopping antibiotics based on PCT values and patterns, in addition to medical history, physical exams, and culture results, could impact the prescription of these drugs. For instance, an observational study that evaluated the effectiveness of a guideline based on serum PCT values showed no difference in mortality compared with the standard care but demonstrated a substantial reduction on the prescription of carbapenems [67].

## 6. Conclusions

Alternative antimicrobials to treat MDR *Acinetobacter baumannii*, *Pseudomonas aeruginosas*, and *Enterobacterales* are probably good options to spare carbapenems and polymyxins, mainly in regions with a lack of resources. However, larger clinical trials are still needed. Additionally, strategies based on cumulative antibiograms and biomarkers, such as PCT, may also optimize the reduction of antimicrobial consumption.

## Figures and Tables

**Figure 1 antibiotics-11-00378-f001:**
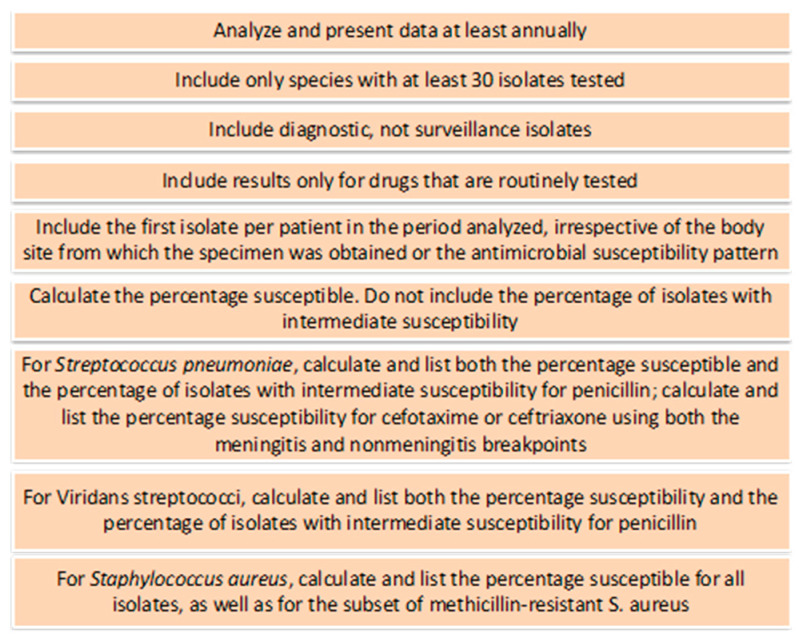
Steps for a cumulative antibiogram (adapted from Hindler JF, Stelling J. Analysis and presentation of cumulative antibiograms: a new consensus guideline from the Clinical and Laboratory Standards Institute. *Clin Infect Dis.* 2007 Mar 15;44(6):867–73).

## Data Availability

Not applicable.

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
