# Peer review of "Antimicrobial Stewardship Programs: A Review of Strategies to Avoid Polymyxins and Carbapenems Misuse in Low Middle-Income Countries"

_antibiotics, 2022, doi:10.3390/antibiotics11030378_

Round 1

Reviewer 1 Report

The manuscript is well written. Only two minor comments:

-in the itroduction th authors should mention the resistance of the Pseudomonas spp and Acinetobacter spp to four ot five (multiple )antibiotics (ECDC (EU/EEA 2019 data)

-Line 86 sulfamethoxazole-trimethoprim (S-T) instead in line 92

Author Response

Thank you for your time to review our manuscript. Additionally, thank for the comments. We have already adjusted it according your suggestions. 

-in the itroduction th authors should mention the resistance of the Pseudomonas spp and Acinetobacter spp to four ot five (multiple )antibiotics (ECDC (EU/EEA 2019 data)

We have included it.

-Line 86 sulfamethoxazole-trimethoprim (S-T) instead in line 92

Thank you for the suggestion!

Reviewer 2 Report

Thank you for the possibility to review: Antimicrobial stewardship programs: a review of strategies to avoid polymyxins and carbapenems misuse in low-middle in-come countries by Fabrício Rodrigues Torres de Carvalho et al.

line 24 please check have been not has been

line 53-56: good scope but please you should resentence 

line 63 scarce please find another word

in the chapter 5 you can explaine something about presepsine and proadreno medullin

And you can add something about the use of microbiome, it is a new possible alternative as adjunct therapy.

These article could provide the riaght way to complete the review

Piccioni A, Saviano A, Cicchinelli S, Valletta F, Santoro MC, de Cunzo T, Zanza C, Longhitano Y, Tullo G, Tilli P, Candelli M, Covino M, Franceschi F. Proadrenomedullin in Sepsis and Septic Shock: A Role in the Emergency Department. Medicina (Kaunas). 2021 Sep 1;57(9):920. doi: 10.3390/medicina57090920. PMID: 34577843; PMCID: PMC8472723.

Zanza C, Romenskaya T, Longhitano Y, Piccolella F, Racca F, Tassi MF, Rubulotta F, Abenavoli L, Shiffer D, Franceschi F, Migneco A, Saviano A, Piccioni A, Ojetti V. Probiotic Bacterial Application in Pediatric Critical Illness as Coadjuvants of Therapy. Medicina (Kaunas). 2021 Jul 30;57(8):781. doi: 10.3390/medicina57080781. PMID: 34440989; PMCID: PMC8399162.

Zanza C, Romenskaya T, Thangathurai D, Ojetti V, Saviano A, Abenavoli L, Robba C, Cammarota G, Franceschi F, Piccioni A, Longhitano Y. Microbiome in Critical illness: An Unconventional and Unknown Ally. Curr Med Chem. 2021 Sep 14. doi: 10.2174/0929867328666210915115056. Epub ahead of print. PMID: 34525908.

Author Response

Thank you for time on out manuscript review. We have followed your suggestions, including the english proof service.  

- line 24 please check have been not has been

- line 53-56: good scope but please you should resentence 

- line 63 scarce please find another word

All of the above were done according the english review editing service. 

- in the chapter 5 you can explain something about presepsine and proadreno medullin

Indeed, those are important biomarkers, specially in the context of sepsis/septic shock. But in this review, we preferred not discuss them because of three mais reasons: they are not so used worldwide at the clinical practice; here is still of a lack of good data about these biomarkers, regarding ASP and antibiotics misuse. But, if you still consider those biomarkers indispensable on this review, we could include a topic about them. Again, thank you very much for your time and suggestions.

Reviewer 3 Report

The manuscript antibiotics-1596791 entitled “Antimicrobial stewardship programs: a review of strategies to avoid polymyxins and carbapenems misuse in low-middle income countries” is an interesting but as per now, incmplete, review about adequate use of antimicrobials in LMICs.

First of all, I would like to comment on some points that refer to the whole manuscript. (1) Bacteria are misspelt in a few cases throughout the manuscript (Pseudomonas aeruginosa –without final ‘s’ or Clostridium difficile with only one ‘l’) (2) Capital letters for antibiotics are not needed (e.g. carbapenem instead of Carbapenem). (3) ASPs are only considered for human health. Anything regarding animal health? As it is a review? Maybe in the discussion? (4) I strongly recommend someone native or with a high level of English to proofread the article.

Specific questions:

-Antimicrobial resistance (AMR) and consumption: consumption of what? Improve the writing and flow of this paragraph (e.g. dots missing after spp and before ‘Additionally’, sentences are not well connected and/or written).

-play a central role? They are important but I would not dare to say central. It has been demonstrated that, in some cases, even with complete withdrawal of antibiotics, resistance to a specific antibiotic has been fixed in a bacterial population. Therefore, ASPs are important but they do not guarantee prevention or reduction of AMR as dynamics can be much more complex than that, especially if they only focus on human health. Use is the most well-known driver of AMR but it is not the only one and only with ASPs we cannot tackle the problem of AMR.

-Different strategies of ASP have been used (e.g. prospective audit and feedback, preauthorization, education program): Sentence is unclear. Suggestion: There are different strategies of ASPs such as prospective audit and feedback, preauthorization or education programs.

-Nevertheless, it is recommended that all core elements are used together to better outcomes (e.g., decreasing AMR and Clostridium  difficile infection): could you better explain what do you mean or imply with this?

-effective in decrease: effective in decreasing

-Additionally, some regions of the globe suffered polymyxin shortage during COVID-19 pandemic with a 400% increase in price (e.g. Brazil).: This is really interesting, any reference or news that we could check?

-You comment about some HICs but the review focuses on LMICs. Contextualise ESBLs in LMIC even if the information is that we lack epidemiological studies or that the information available is scarce because I was confused to read about Belgium, France, Germany and Ireland but not any LMIC.

-Section 2 and 3 are unstructured and information is mixed-up, even difficult to understand sometimes. I would recommend clearly following a plot or points to consider in each section.

-Section 4 is interesting and the cumulative antibiogram seems useful. But the reference and picture mention an article from 2007. If this was depicted then, why is it not well adopted? Where is the assessment of doing that in LMICs and the problems that LMICs are facing if they want to adopt this approach?

-Section 5 is missing context both of the CPR and PCT, missing details and general information, although later on it focuses on PCT. Pros an cons of using one or the other (e.g, time, sensitivity, cost, infrastructure…). CPR seems more used than PCT, why? If PCT seems more specific according to the information provided? 

-Alternative antimicrobials to treat MDR…: General and standard sentence in the conclusion…which ones? Specific conclusions are needed as this does not seem a sentence to conclude.

To finish, I would like to assess this manuscript which (i) is missing relevant information and specific details for LMICs, (ii) must be proofread and (iii) needs a better structure and writing. I must recommend “MAJOR REVISIONS”.

Author Response

Dear reviewer, thank you for your time on the revision of our manuscript. Indeed, your commentaries improved the quality of our paper. Bellow, we have answered your commentaries. All the suggested changes are included in the new version of the manuscript (please, see the file attached).

-Antimicrobial resistance (AMR) and consumption: consumption of what? Improve the writing and flow of this paragraph (e.g. dots missing after spp and before ‘Additionally’, sentences are not well connected and/or written).

We have sent to an english proof service. Thank you.

-play a central role? They are important but I would not dare to say central. It has been demonstrated that, in some cases, even with complete withdrawal of antibiotics, resistance to a specific antibiotic has been fixed in a bacterial population. Therefore, ASPs are important but they do not guarantee prevention or reduction of AMR as dynamics can be much more complex than that, especially if they only focus on human health. Use is the most well-known driver of AMR but it is not the only one and only with ASPs we cannot tackle the problem of AMR.

Indeed, we also agree with it. We have improved it. 

-Different strategies of ASP have been used (e.g. prospective audit and feedback, preauthorization, education program): Sentence is unclear. Suggestion: There are different strategies of ASPs such as prospective audit and feedback, preauthorization or education programs.

Thank you, we have corrected it. 

-Nevertheless, it is recommended that all core elements are used together to better outcomes (e.g., decreasing AMR and Clostridium  difficile infection): could you better explain what do you mean or imply with this?

Indeed, we should have clarified it. Thank you.

-effective in decrease: effective in decreasing

We corrected it.

-Additionally, some regions of the globe suffered polymyxin shortage during COVID-19 pandemic with a 400% increase in price (e.g. Brazil).: This is really interesting, any reference or news that we could check?

Indeed, we have struggled during the shortage. References are included - one from Journal of Hospital Infection and other from one the majors Brazilian newspapers (i.e., O Globo).

-You comment about some HICs but the review focuses on LMICs. Contextualise ESBLs in LMIC even if the information is that we lack epidemiological studies or that the information available is scarce because I was confused to read about Belgium, France, Germany and Ireland but not any LMIC.

Thank you, we have included references from LMIC.

-Section 2 and 3 are unstructured and information is mixed-up, even difficult to understand sometimes. I would recommend clearly following a plot or points to consider in each section.

Indeed, we have pointed each section according the sub-sections theme. Thank you.

-Section 4 is interesting and the cumulative antibiogram seems useful. But the reference and picture mention an article from 2007. If this was depicted then, why is it not well adopted? Where is the assessment of doing that in LMICs and the problems that LMICs are facing if they want to adopt this approach?

Thank you, we have approached it. Indeed an important issue.

-Section 5 is missing context both of the CPR and PCT, missing details and general information, although later on it focuses on PCT. Pros and cons of using one or the other (e.g, time, sensitivity, cost, infrastructure…). CPR seems more used than PCT, why? If PCT seems more specific according to the information provided? 

Thank you for the observations! We have improved the text of this section. 

-Alternative antimicrobials to treat MDR…: General and standard sentence in the conclusion…which ones? Specific conclusions are needed as this does not seem a sentence to conclude.

Thank you, we have improved it.

Round 2

Reviewer 2 Report

It Is nicer now

Reviewer 3 Report

The manuscript is now much better than before, much more complete and clearer. After addressing all my previous concerns and comments, I recommend accepting the manuscript for publication. However, there are still some spelling errors in the manuscript and I would consider to proofread the language before is publicly available (e.g. surveillance, Clostridioides difficile).